# Longitudinal Analysis Evaluating Self-Reported CPAP Use for OSA during the COVID-19 Pandemic

**DOI:** 10.3390/brainsci12020131

**Published:** 2022-01-19

**Authors:** Taylor Torrence Teague, Ahmad Debian, Manasa Kokonda, Sonal Malhotra, Emily Arentson-Lantz, Fidaa Shaib, Sara Nowakowski

**Affiliations:** 1Department of Medicine, Baylor College of Medicine, Houston, TX 77030, USA; taylor.torrenceteague@bcm.edu (T.T.T.); ahmad.debian@bcm.edu (A.D.); manasa.kokonda@bcm.edu (M.K.); sonal.malhotra@bcm.edu (S.M.); shaib@bcm.edu (F.S.); 2Department of Pediatrics, Texas Children’s Hospital, Houston, TX 77030, USA; 3Center for Innovations in Quality, Effectiveness and Safety, Michael E. DeBakey Veteran Affairs Medical Center, Houston, TX 77021, USA; 4Department of Nutrition & Metabolism, University of Texas Medical Branch, Galveston, TX 77555, USA; ejlantz@utmb.edu

**Keywords:** obstructive sleep apnea, OSA, continuous positive airway pressure, CPAP, adherence, COVID-19 pandemic, SARS-CoV-2

## Abstract

Continuous positive airway pressure therapy (CPAP) is a highly effective treatment for obstructive sleep apnea (OSA), but CPAP adherence remains suboptimal. The COVID-19 pandemic significantly altered sleep medicine services and aspects of daily living for sleep medicine patients, which may further compromise CPAP adherence. Sleep medicine patients were distributed an online survey at baseline and six months later (January–May 2021). Participants answered questions regarding CPAP use (any changes in CPAP use, sleep quality with CPAP use, CPAP use as advised, and changes in daily habits). Eighty-one adults completed the baseline survey, and 54 adults completed the follow-up survey. Twenty-seven participants reported a diagnosis of OSA and were prescribed CPAP (mean age 58 ± 18.2 years, 48% female, 67% Caucasian). Longitudinal analysis with chi-square association testing showed significant changes in CPAP use as advised and significant improvements in sleep quality with CPAP use when comparing the baseline to six-month follow-up survey. Additionally, logistic regression was performed to determine if pre-pandemic sleep study results (apnea-hypopnea index and respiratory disturbance index) predicted self-reported CPAP use during the pandemic, though no association was found. Throughout the pandemic, sleep medicine patients improved their CPAP use as advised and reported significant improvements in sleep quality with CPAP use.

## 1. Introduction

Obstructive sleep apnea (OSA) is a common chronic disorder that affects 3–9% of the population [1]. It is characterized by upper airway collapse (either partial or complete) during sleep that leads to sleep fragmentation and episodes of oxyhemoglobin desaturation. Continuous positive airway pressure therapy (CPAP) is a highly effective treatment for OSA since it eliminates airway collapse during sleep [2,3,4], however adherence to CPAP remains suboptimal (adherence rates range from 30–60% [5,6,7,8,9,10]). CPAP adherence has been shown to improve with certain interventions, and the American Academy of Sleep Medicine (AASM) recommends educational, behavioral, and troubleshooting interventions for adults with OSA [11]. Despite decades of research, there is no single factor that reliably predicts CPAP adherence (or non-adherence). Numerous factors seem to contribute to adherence, including patient characteristics (age, medical comorbidities, etc.), disease severity (defined by apnea-hypopnea index (AHI) and oxyhemoglobin desaturation), and both clinical and educational support [12,13,14]. Some have investigated predictors of CPAP non-adherence specifically during the COVID-19 pandemic and have similarly observed that multiple factors seem to predict CPAP non-adherence: sex, age, and disease severity [15].

The COVID-19 pandemic that spread throughout the United States in early 2020 significantly altered sleep medicine services and interactions with sleep medicine patients. Initially, many sleep centers that performed diagnostic polysomnography testing completely closed. Others remained open after implementing procedural changes but functioned at reduced capacity, ultimately leading to a delay in diagnosis and treatment. In-person appointments were switched to telemedicine, both video and telephone. These changes could have impacted patients’ CPAP adherence rates. For example, in-person visits offer a hands-on approach to CPAP troubleshooting, whether it is related to the CPAP device itself (e.g., humidity settings, ramp settings, etc.) or its supplies (e.g., mask discomfort, mask leak, etc.). Furthermore, at the beginning of the outbreak, there were opposing recommendations regarding CPAP use among patients with OSA. Some supported cessation of CPAP for patients with OSA because of the risk for CPAP-induced aerosolization of SARS-CoV-2 [16], while others encouraged CPAP continuation after considering the harm to benefit ratio [17]. This discrepancy in recommendations could have led to confusion among patients with OSA and contributed to suboptimal CPAP adherence rates. It is also important to note that the pandemic continues to impact multiple facets of daily living for sleep medicine patients: work schedule, personal finances, physical activity, and mental and physical health, all of which can result in additional stress that negatively affects sleep and possibly contributes to CPAP non-adherence.

Given these circumstances, we hypothesized that the COVID-19 pandemic further compromised CPAP adherence for sleep medicine clinic patients. This longitudinal analysis aimed to describe self-reported CPAP use in sleep medicine clinic patients during the COVID-19 pandemic and determine if any pre-pandemic sleep study results were associated with CPAP use over a six-month period.

## 2. Materials and Methods

### 2.1. Participants

Sleep medicine clinic patients from Baylor College of Medicine (BCM) at the McNair Campus were distributed an online survey at baseline (June–November 2020) followed by a 6-month follow-up survey (January–May 2021). Research participants were identified by electronic health record (EHR) chart review. To be considered eligible, participants had to be a sleep medicine patient at BCM, at least 18 years old, and English speaking. There were no exclusion criteria. Once identified, eligible participants received an email EHR message that described the study. After agreeing to participate in the study, participants then received a link to an online Research Electronic Data Capture (REDCap) survey.

Data were obtained through an online self-reported REDCap survey and through chart review. The REDCap survey included a questionnaire that contained sociodemographic characteristics, professional information, CPAP use (additional details below), validated Patient-Reported Outcomes Measurement Information System (PROMIS) sleep disturbance, PROMIS sleep impairment, PROMIS severity of substance use, PROMIS alcohol use, PROMIS depression, PROMIS emotional support, PROMIS social isolation, PROMIS informational support, PROMIS smoking coping expectancies for all smokers, PROMIS anxiety, perceived stress scale, insomnia severity index, Coronavirus anxiety scale, and brief-COPE (Coping Orientation to Problems Experienced) [18,19,20,21,22,23,24,25,26]. PROMIS measures were identical on baseline and 6-month follow-up survey. Patients were instructed to think about the past month when answering questions. Through chart review, we gathered patient information regarding baseline diagnostic sleep study results, medical history, surgical history, medications, and demographics.

In both the baseline and 6-month questionnaires, participants specified if they currently had a positive airway pressure (PAP) device. For those that answered yes, additional details were obtained: any changes in CPAP usage per night (defined as 1–3 h per night or ≥4 h per night), any changes in CPAP usage per week (defined as 1–3 nights per week or ≥4 nights per week), any changes to sleep quality using CPAP during COVID-19 pandemic, any changes to CPAP therapy as advised during the COVID-19 pandemic and any changes in their daily habits. CPAP therapy as advised was defined as using CPAP for entirety of participants’ overnight sleep. We did not define “optimal CPAP use”, though many have adopted the United States Medicare policy [27] for the continued coverage of a PAP device, which is defined as CPAP use for ≥4 h per night on at least 70% of nights during a consecutive 30-day period during the first 3 months of initial usage.

No specified interventions were performed (i.e., patient education regarding CPAP use during the COVID-19 pandemic) between the baseline and 6-month follow up surveys.

The study was approved by the Baylor College of Medicine Institutional Review Board. Informed consent was obtained from all subjects involved in the study.

### 2.2. Data, Analysis, and Statistics

Longitudinal analysis with chi-square association testing evaluated for any significant changes in CPAP use at baseline survey (conducted throughout the initial COVID-19 surge in Texas) to 6-month follow-up survey. Chi-square association testing was also performed to determine if changes in daily habits during the COVID-19 pandemic predicted changes in CPAP use. Logistic regression was performed using Statistical Analysis System (SAS) to determine if any pre-pandemic sleep study results (specifically AHI and respiratory disturbance index (RDI)) were associated with CPAP use during the COVID-19 pandemic.

## 3. Results

### 3.1. Characteristics of Participants

A total of 571 eligible participants were identified. Eighty-one adults completed the baseline survey and 54 adults completed the six-month follow-up survey, as depicted in Figure 1. The mean age of participants who completed the baseline survey was 54.8 ± 15.9 years, with 56% female and 69% Caucasian (as seen in Table 1). In the follow-up survey, the mean age of participants was 55.2 ± 18.4 years, with 66% female and 70% Caucasian.

Twenty-seven participants reported a diagnosis of OSA and were prescribed CPAP. These participants were followed throughout the pandemic. Their mean age was 58 ± 18.2 years, with 48% female, and 67% Caucasian. Mean body mass index (BMI) was 34.9 ± 12.4 kg/m^2^. Chart review indicated that all 27 participants had a diagnostic sleep study prior to March 2020, with a range from 2015 to February 2020. Baseline AHI on diagnostic sleep study was 8.2 ± 10.7 events/hour. The vast majority of participants who utilized CPAP for OSA reported using it for ≥4 h per night (96%) and for ≥4 days per week (100%).

### 3.2. Longitudial Analysis of CPAP Use during the COVID-19 Pandemic

Longitudinal analysis with chi-square association testing showed significant changes in CPAP use as advised and significant improvements in sleep quality with CPAP use when comparing baseline survey conducted early in the pandemic to a six-month follow-up survey, as seen in Table 2.

### 3.3. Analysis of Sleep Study Results and CPAP Use during the COVID-19 Pandemic

Logistic regression was performed using SAS to determine if pre-pandemic sleep study results (specifically AHI and RDI) were associated with changes in CPAP use during the COVID-19 pandemic. No association was found, as seen in Table 3.

### 3.4. Analysis of Changes in Daily Habits during the COVID-19 Pandemic and CPAP Use

Chi-square association testing was performed using SAS to determine if changes in daily habits during the COVID-19 pandemic were associated with changes in CPAP use. No association was found, as seen in Table 4.

## 4. Discussion

In our longitudinal analysis, sleep medicine clinic patients reported an improvement in their CPAP use as advised and reported significant improvements in their sleep quality with CPAP use. Our observations of improved adherence during the COVID-19 pandemic are substantiated by others [15,28]. Attias et al. observed a 3.9% increase in adherence (from a mean value of 386 min per night pre-COVID-19 to 401 min per night during lockdown) in 7485 patients with OSA [28]. Demirovic et al. found that CPAP adherence improved during the COVID-19 pandemic in patients with severe OSA, in those with optimal pre-lockdown CPAP adherence (CPAP usage for ≥4 h/night on at least 70% of nights), women, and those younger than 58 years old [15].

It is important to emphasize that our patients had optimal CPAP adherence (assuming Medicare’s definition [27]) at baseline and again at six-month follow-up: 96% reported CPAP usage of ≥4 h and 100% reported CPAP usage of ≥4 nights per week. However, we observed a significant improvement in CPAP use as advised, suggesting that patients extended their CPAP usage for the entirety of their overnight sleep (rather than limit usage to minimum of 4 h per night). This could potentially explain why patients also reported significant improvements in their sleep quality with CPAP use throughout the course of the pandemic. When patients extend their CPAP usage for the entirety of their overnight sleep, there will be fewer episodes of upper airway collapse, leading to less sleep fragmentation, improved oxyhemoglobin saturation, and theoretically overall better sleep quality.

Unlike previous studies, we did not observe any association between pre-pandemic factors, such as pre-pandemic sleep study AHI and RDI, and CPAP use during the COVID-19 pandemic. Others like Demirovic et al. recognized that male sex, older age, and severity of OSA (specifically severe OSA) were less likely to improve CPAP usage during the lockdown. Our results are not surprising though and likely because our patients’ self- reported optimal CPAP adherence (96% reported using it for ≥4 h per night and 100% reported using it for ≥4 days per week). If we had identified an association between pre-pandemic sleep study results and CPAP use during COVID-19 pandemic, then this observation could have had potential clinical significance. Adherence to CPAP is notoriously low with adherence rates ranging from 30% to 60% [5,6,7,8,9,10]. The AASM recommends certain interventions (educational, behavioral, and troubleshooting interventions) to improve PAP adherence in adults with OSA [11]. Perhaps these interventions could be targeted and tailored to patients with risk factors of CPAP non-adherence.

Our study is unique because it is the first longitudinal analysis that describes self-reported CPAP use throughout the course of the COVID-19 pandemic in the United States. Other studies have evaluated CPAP adherence at only a single period during the COVID-19 pandemic [15,28]. A recent cross-sectional study conducted by Tepwimonpetkun et al. evaluated PAP adherence during the COVID-19 pandemic (between February and October 2020) in patients from Bangkok, Thailand [29]. Tepwimonpetkun et al. observed that self-reported PAP adherence increased in a subgroup of their 156 patients. However, according to objective data (i.e., PAP download), there was no significant increase in PAP usage during the COVID-19 pandemic. Our baseline survey occurred from June to November 2020 (throughout Texas’s initial surge and early into its second surge) when the seven-day average of COVID-19 cases ranged from 1370 to 11,158 [30]. Our six-month follow-up survey occurred from January to May 2021 (approaching the peak of the second surge and into the nadir following the second surge) when the seven-day average of COVID-19 cases ranged from 17,859 to a nadir of 1217 [30]. The timing of our questionnaires could be considered a limitation by some since they did not occur at a comparable severity during the various COVID-19 surges. However, our aim was to describe self-reported CPAP use throughout the COVID-19 pandemic at pre-specified times (baseline then six months later), and unfortunately we did not have the ability to time our questionnaires with the different COVID-19 surges in Texas.

Our study also highlights the importance of utilizing CPAP as advised (i.e., for the entirety of patients’ overnight sleep rather than limiting its use to 4 h per night) as this could have contributed to our observation that patients reported significant improvements in their sleep quality with CPAP use.

Our study has limitations. First, our study has a small sample size with 81 patients completing our baseline survey and 54 patients completing our follow-up survey. Additionally, only 27 participants utilized CPAP therapy for their underlying OSA. Furthermore, two thirds of our participants were of Caucasian race. Altogether, these factors limit our study’s generalizability. It is important to note that 66% of responders in our follow-up survey were women, which is surprising given the high prevalence of OSA in men. However, this likely did not impact our study’s generalizability, because the 27 participants utilizing CPAP were well balanced with 52% men and 48% women. Second, our study is a convenience sample of sleep medicine clinic patients, so there may be a bias regarding those who chose to complete the survey. For example, only those capable of using electronic mail and EHR messaging were able to participate in our study. Moreover, we offered no monetary reimbursement for completing our surveys, so our patient population is likely highly motivated in general. Thus, our results could be biased due to high motivation regarding CPAP use, too. Third, the results of our longitudinal analysis are based upon self-reported CPAP use with no objective data (i.e., CPAP download data) to corroborate participants’ responses.

## 5. Conclusions

Throughout the course of the COVID-19 pandemic, sleep medicine clinic patients reported improvement in their CPAP use as advised and reported significant improvements in their sleep quality with CPAP use. Furthermore, in our study population, pre-pandemic factors (specifically sleep study results like AHI and RDI) were not associated with CPAP non-adherence. Additionally, no association was identified between changes in daily habits during the pandemic and CPAP use. Additional studies are needed to evaluate modifiable predictors of CPAP adherence in the hope of improving CPAP adherence in patients with OSA.

## Figures and Tables

**Figure 1 brainsci-12-00131-f001:**
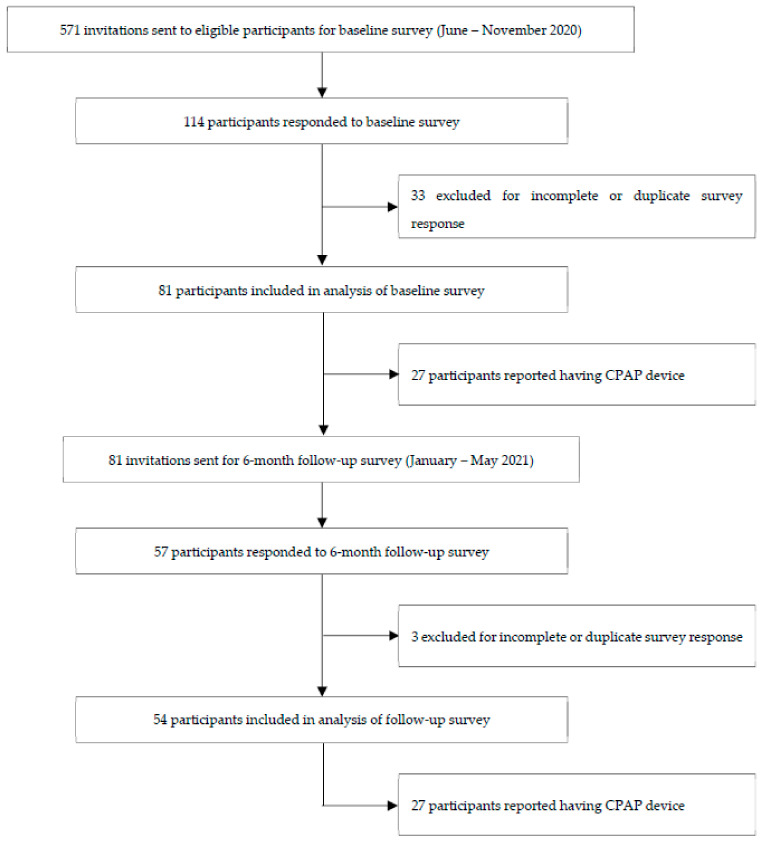
Flow diagram of participant selection.

**Table 1 brainsci-12-00131-t001:** Characteristics of participants.

Demographic Data	Participants in Baseline Survey, *n* = 81	Participants in 6-Month Follow-Up Survey, *n* = 54	Participants with CPAP Device, *n* = 27
Age, years	54.8 ± 15.9	55.2 ± 18.4	58 ± 18.2
Sex			
Female	45 (55.6)	33 (66.1)	13 (48.2)
Male	36 (44.4)	21 (38.9)	14 (51.8)
Race			
Caucasian	56 (69.1)	38 (70.4)	18 (66.7)
African American	16 (19.8)	12 (22.2)	7 (25.9)
Asian	4 (4.9)	3 (5.6)	1 (3.7)
Not reported	5 (6.2)	1 (1.9)	1 (3.7)
^α^ BMI, kg/m^2^			34.9 ± 12.4
^β^ AHI, events/hour			8.2 ± 10.7
^Ω^ Current smoker			1 (3.9)
^ϕ^ Diabetes mellitus			10 (38.5)
Self-reported CPAP usage per night, hours			
1–3 h			1 (3.7)
≥4 h			26 (96.3)
Self-reported CPAP usage per week, nights			
1–3 nights			
≥4 nights			27 (100)

Note: Data is presented as mean ± standard deviation or *n* (%). ^α^ Missing BMI from 2 patients (*n* = 25). ^β^ Missing AHI from 3 patients (*n* = 24). ^Ω^ Missing 1 patient (*n* = 26). ^ϕ^ Missing 1 patient (*n* = 26). Abbreviations: CPAP, continuous positive airway pressure. BMI, body mass index. AHI, apnea-hypopnea index.

**Table 2 brainsci-12-00131-t002:** Longitudinal analysis of CPAP use during COVID-19 pandemic using chi-square association testing.

Variable	Description	Participants in Baseline Survey, *n* (%)	Participants in 6-Month Follow-Up Survey, *n* (%)	*p* Value
CPAP use	Yes	27 (33)	27 (50)	-
	No	54 (67)	27 (50)	
Change in CPAP use	No change	23 (85.2)	24 (88.9)	
	Use more	3 (11.1)	2 (7.4)	0.166
	Use less	1 (3.7)	1 (3.7)	
Change in sleep quality with CPAP use	No change	9 (33.3)	7 (25.9)	
	Better	13 (48.2)	19 (70.4)	0.012 *
	Worse	5 (18.5)	1 (3.7)	
Change in CPAP use as advised	Unsure	6 (22.2)	0	
	No	4 (14.8)	3 (11.1)	0.003 *
	Yes	17 (63.0)	24 (88.9)	

Abbreviations: CPAP, continuous positive airway pressure. * Indicates statistical significance. *p* value calculated with chi-square association test for categorical variables and used exact testing when needed.

**Table 3 brainsci-12-00131-t003:** Point-estimate of sleep study results and CPAP use during the COVID-19 pandemic.

Sleep Study Results	Change in CPAP Use	Sleep Quality with CPAP Use	Change in CPAP Use as Advised
AHI	1.02 (0.65)	0.87 (0.13)	1.03 (0.61)
RDI	9.53 (0.75)	0.11 (0.74)	0.91 (0.51)

Note: Data is presented as point-estimate (*p*-value). Abbreviations: CPAP, continuous positive airway pressure. AHI, apnea-hypopnea index. RDI, respiratory disturbance index.

**Table 4 brainsci-12-00131-t004:** Analysis of changes in daily habits during the COVID-19 pandemic and CPAP use.

Daily Habit	Description	No Change in CPAP Use, *n* (%)	More CPAP Use, *n* (%)	Less CPAP Use, *n* (%)	*p* Value
Employment change	Yes	4 (14.8)	-	-	0.605
	No	20 (74.1)	2 (7.4)	1 (3.7)	
Healthcare change	Yes	15 (55.6)	2 (7.4)	-	0.155
	No	9 (33.3)	-	1 (3.7)	
Electronics	Less time	1 (3.7)	-	-	0.331
	More time	16 (59.3)	2 (7.4)	1 (3.7)	
	No change	7 (25.9)	-	-	
Change in sleep medication	Yes	1 (3.7)	-	-	0.074
	No	21 (91.3)	1 (3.7)	-	
	No medications	2 (7.4)	1 (3.7)	1 (3.7)	
Exercise	Less time	13 (48.2)	-	1 (3.7)	0.072
	More time	3 (11.1)	-	-	
	No change	8 (29.6)	2 (7.4)	-	
Sunlight exposure	Less time	15 (55.6)	1 (3.7)	1 (3.7)	0.124
	More time	3 (11.1)	1 (3.7)	-	
	No change	6 (22.2)	-	-	
Caffeine consumption	Less	-	-	-	0.132
	More	4 (14.8)	-	1 (3.7)	
	No change	20 (74.1)	2 (7.4)	-	

Abbreviations: CPAP, continuous positive airway pressure.

## Data Availability

The data presented in this study are available on request from the corresponding author. The data are not publicly available due to privacy protection reasons.

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
