# Peer review of "Longitudinal Analysis Evaluating Self-Reported CPAP Use for OSA during the COVID-19 Pandemic"

_brainsci, 2022, doi:10.3390/brainsci12020131_

Round 1

Reviewer 1 Report

This is cross sectional study that evaluates the CPAP compliance during COVID 19 pandemic. Sleep medicine clinic patients were distributed an online survey at baseline and 6 months later. Participants answered questions regarding CPAP use, specifically any changes in CPAP use, sleep quality with CPAP use, CPAP use as advised, and changes in daily habits. Logistic regression was performed using SAS to determine if pre-pandemic sleep study results (AHI and RDI) predicted self-reported CPAP use during the pandemic. Eighty-one adults completed baseline survey, and 54 adults completed follow-up survey among which twenty-seven participants reported a diagnosis of OSA and were prescribed CPAP. Longitudinal analysis with chi-square association testing showed significant changes in CPAP use as advised and significant improvements in sleep quality with CPAP use when comparing baseline to 6-month follow-up survey. Authors concluded that throughout the course of the pandemic, sleep medicine clinic patients improved their CPAP use as advised and reported significant improvements in their sleep quality with CPAP.

I have some comments and suggestions for revision to propose to the authors

  1. Title: I suggest authors to consider adding in the title subjective assessment or self-reported e. Longitudinal Analysis Evaluating subjective assessment of CPAP Use during the COVID-19 Pandemic.
  2. Methods: there are some information’s needed:

Research participants were identified by electronic health record (EHR) chart review as patients previously seen in BCM sleep medicine clinic. How many records are in HER, do you have estimation of response rate? Where there any specific or predetermined inclusion and exclusion criteria for study participants?

How did authors define CPAP usage for those 27 participants who were prescribed with CPAP therapy, or more precisely did authors define optimal CPAP usage? What about definition of more or less use of CPAP, how was this defined in terms of minutes, hours per day?

Were all of 27 CPAP patient’s optimal users et baseline? What about information on COVID-19 infection prior to or during study or during 6 month follow up? Where there any data collected regarding COVID-19 infection for study participants which might have effect on results.

Also, for how long study participants were treated for OSA? Was there a prespecified time of including patients under CPAP, or how long before patients initiated to the study? Patients with a longer OSA diagnosis could have been better educated in terms of CPAP adherence. Where there any interventions performed during 6 months follow up in terms of educations or advises for CPAP users due to COVID-19 pandemic.

You pointed out collection of medical history data and medications of study participants. You might consider adding more of those information’s that you collected about study participants in Table 1, or at least in population of CPAP users. Did you analyse effect of potential comorbidities among CPAP users such as arterial hypertension, diabetes on CPAP adherence?

Have authors calculated the sample size for the power analysis of their study, which specific primary endpoint (-s) was this analysis based on.

Finally, I strongly recommend to put in conclusion as well as in the title of the study term “subjective assessment or self reported” because all conclusions of the study are based on subjective assessment which is the one of limitation of the study and at some point might be influenced by the motivation of chosen study participants.

Reviewer 2 Report

ABSTRACT: Well written

INTRODUCTION: Acceptable, although might be improved with better rationalization.

PARTICIPANTS: Small sample size

Correction that have to be made: Each abbreviation needs to have a full name when first mentioned in text. Please add as follows:

Please ad a full name for abbreviation REDCap survey

Please ad a full name for abbreviation PROMIS

Please ad a full name for abbreviation brief-COPE

Please ad a full name for abbreviation SAS

RESULTS: There are some corrections that have to be made in order to accept this paper. Some cardinal mistakes are present, and they have to be corrected.

It is necessary to provide an additional table with demographic characteristic of your patients (besides age and gender, give us information about BMI, smoking status, AHI, time of OSA diagnosis at baseline)

Crucial mistake has been made wit he repetition of data in Table 2 and Figure 1. Showing the same result in two forms is never acceptable in scientific papers, thus Figure 1 is redundant, and you should not present it in this paper.  

DISCUSSION: Please ad discussion on the selection of patients since only patients educated in electronic mail were able to participate; Please discuss that 66% of participants were female patients, although there is significantly more man in OSA population; Is your sample representative? Also, please discuss  how motivation of participants to be a part of the study influenced results.

Reviewer 3 Report

In this work, Teague and colleagues assess the use of continuous positive airway pressure (CPAP) in obstructive sleep apnea (OSA) patients during two different phases of the pandemic and they show that the usage is increased at 6 months compared to baseline. This study has several limitations that preclude further assessment, see as follows:

  1. The authors do not support their aims with solid rationales.
    1. They decide to record a baseline questionnaire in December, amidst the course of the second wave in Texas, which is hardly representative of any specific hallmark in the pandemic (it was not the beginning of the wave, but cases were still increasing, though not at the peak), and compare it to another questionnaire recorded in May, when the second wave’s tail was long gone (so, after more than two months with essentially non/little Covid). Thus, these two points in time capture non-specific data and are poorly comparable for characteristics. It follows that the authors’ assumption that the pandemic increases CPAP usage is too polluted by the aforementioned factors to be conclusive.
    2. The authors try to find predictors for CPAP use during the pandemic. This comes “out of the blue” in the abstract and in the methods, as it is not adequately justified by 1) enough literature background in the introduction, and 2) a statistical power analysis (27 patients really seem too small of a sample size).
  2. The method section is missing too many key details: in particular, inclusion/exclusion criteria are missing (it is really not clear why the authors recruited also patients without OSA), “chart review” does not explain how sleep quality and CPAP use (what was the used unit?) were quantified (also, “CPAP use as advised” should be the same for every patient, i.e. 4 h/night), and “change in CPAP use” really seem the same thing of “change in CPAP use as advised”.
  3. The authors should describe better what this study adds to the field, since (as also the authors mention), other studies have highlighted an improvement of CPAP use/adherence during the pandemic/lockdowns.

In addition I have other minor, although important, comments:

  1. The title should specify that the CPAP use is for OSA.
  2. In the abstract and the main body, the authors should spell out all acronyms.
  3. In the abstract, it is not clear what changes in CPAP use were evident.
  4. Collapse is a better word than closure.
  5. Citation number 2 seems to be more related to CPAP compliance than to what it is referenced to.
  6. It seems that lines 40-41 need an update.
  7. The authors should specify, if true, that all PROMIS items were questionnaires, and that, if true, questionnaires were identical at baseline and after 6 months.
